# MetaHiC phage-bacteria infection network reveals active cycling phages of the healthy human gut

Martial Marbouty[1]*, Agnès Thierry[1], Gaël A Millot[2], Romain Koszul[1]*

[1]Institut Pasteur, Unité Régulation Spatiale des Génomes, CNRS, UMR 3525, Paris, France; [2]Institut Pasteur, Bioinformatics and Biostatistics Hub, CNRS, USR 3756, Paris, France

**Abstract** Bacteriophages play important roles in regulating the intestinal human microbiota composition, dynamics, and homeostasis, and characterizing their bacterial hosts is needed to understand their impact. We applied a metagenomic Hi-C approach on 10 healthy human gut samples to unveil a large infection network encompassing more than 6000 interactions bridging a metagenomic assembled genomes (MAGs) and a phage sequence, allowing to study in situ phage-host ratio. Whereas three-quarters of these sequences likely correspond to dormant prophages, 5% exhibit a much higher coverage than their associated MAG, representing potentially actively replicating phages. We detected 17 sequences of members of the crAss-like phage family, whose hosts diversity remained until recently relatively elusive. For each of them, a unique bacterial host was identified, all belonging to different genus of Bacteroidetes. Therefore, metaHiC deciphers infection network of microbial population with a high specificity paving the way to dynamic analysis of mobile genetic elements in complex ecosystems.

*For correspondence:
martial.marbouty@pasteur.fr
(MM);
romain.koszul@pasteur.fr (RK)

**Competing interests:** The authors declare that no competing interests exist.

## Introduction

The gut hosts a complex microbial ecosystem composed of bacteria, archaea, eukaryotic microorganisms, and viruses. High-throughput sequencing allows in-depth investigation of the gut microbiota (*Arumugam et al., 2011*; *Li et al., 2014*), unveiling links between the balance of this complex ecosystem with a wide range of human diseases (*Cho and Blaser, 2012*), including autoimmunity (*Wu and Wu, 2012*) or neurological disorders (*Cryan and Dinan, 2012*). The regulation of the human gut microbiome is therefore much investigated, notably the role played by mobile genetic elements, and especially the collection of viruses present in the gut (or virome), in influencing its homeostasis maintenance. Indeed, the virome can either directly influence the immune system or reshape the bacterial component of the gut (*Chatterjee and Duerkop, 2018*; *Gogokhia et al., 2019*; *Keen and Dantas, 2018*; *Norman et al., 2015*). The gut virome is dominated by temperate bacteriophages belonging to the Caudovirales family (*Minot et al., 2011*; *Shkoporov et al., 2019*), but how the microbiome and the virome influence each other over time remains largely unknown, with only a few studies characterizing concomitantly host-phage relationships in individual gut sample (*Džunková et al., 2019*; *Shkoporov et al., 2018*). Reaching at a better characterization of these relationships is nevertheless a key, primary step to fully understand the impact of phages on microbial population (*Edwards et al., 2016*), hence our ability to influence it for biomedical applications. The ratio between phages and bacteria in the gut is also a relevant metric to understand further the regulation of the microbiota equilibrium. This ratio is typically assessed from the quantification of virus-like particles (VLP), corresponding to free phages produced during lytic cycle (*Kleiner et al., 2015*), with respect to the amount of bacteria cells. First evidences indicated ratios between 1 and 0.001 (*Hoyles et al., 2014*; *Kim et al., 2011*; *Shkoporov et al., 2019*), suggesting that the intestinal

tract is quite different from other ecosystems such as oceans where phages largely outnumber bacteria (*Roux et al., 2016*). There is therefore a general need for comprehensive studies aiming at characterizing (1) which phage infects which bacterial species and (2) the relative phage and bacterial concentrations for each phage-host pair in the human gut.

Several methods have been developed in recent years to capture the global host-phage network directly in their natural environment: proximity ligation (*Marbouty et al., 2014*; *Marbouty et al., 2017*; *Stalder et al., 2019*; *Yaffe and Relman, 2020*), viral tagging (*Deng et al., 2014*; *Džunková et al., 2019*), or single amplified genomes (*Labonté et al., 2015*; *Munson-McGee et al., 2018*). We notably showed that meta3C, a proximity ligation-based approach, allows the assembly and scaffolding of dozens of nearly complete or complete phages and bacteria genomes, as well as their relationships, when applied on a single mammalian gut sample (*Marbouty et al., 2017*). These genomes and the corresponding infection network are inferred from the quantification of collision frequencies between DNA segments in the population, captured using a derivative of chromosome conformation capture (*Dekker et al., 2002*).

In the present work, we developed and apply an improved meta3C protocol (metaHiC) to reconstruct the large landscape of phages and their interacting bacteria in 10 individual, healthy human gut microbiota. By capturing and quantifying the DNA-DNA collisions during phages replication inside their bacterial hosts, we characterized a total of 6,763 unique host-phage contigs pairing, which represent, to our knowledge, the largest infection network of the human gut microbiome to date. While three-quarters of the detected phages appeared to be dormant lysogenic phages, a significant proportion (~5%) of the phage's population exhibits a sequencing read coverage higher than the one of their corresponding hosts, suggesting these phages are actively dividing. We also characterized different crAss-like phages candidates as well as their relatively elusive bacterial hosts, all belonging to Bacteroidales order. Through the use of such approach to determine phage's host target in single samples, this study can have practical consequences, notably for fecal microbiota transplantation (*Chehoud et al., 2016*) or phage therapy (*Dufour et al., 2019*; *Kortright et al., 2019*).

## Results

### High-quality proximity ligation meta3C / metaHiC libraries of human gut microbiome

Frozen stool samples from 10 healthy adults from the Institut Pasteur Biobanque (agreement N18) (*Figure 1a*) were processed with the established meta3C protocol using either HpaII or MluCI as a restriction enzyme (*Marbouty et al., 2017*). Each meta3C library was sequenced (between 43 and 206 million paired-end [PE] reads) and assembled separately using MEGAHIT (*Li et al., 2015*), resulting in 10 human gut assemblies ranging from 242 to 617 Mb (total size = 3.45 Gb for 1,485,156 contigs) ('Materials and methods') (*Supplementary file 1*). To assess the amount of useful, long-range 3D DNA contacts in the different meta3C libraries, each read of a pair was aligned against its corresponding assembly. The number of pairs for which both reads aligned unambiguously on two different contigs (mapping quality $\geq$20) was used to compute a '3D ratio', by dividing it with the number of pairs that aligned unambiguously within individual contigs (*Figure 1—figure supplement 1a* and *Supplementary file 2*). 3D ratios using the standard meta3C protocol range from 1.92% to 14.58%, a variability that results most likely from samples heterogeneities and from the restriction enzymes used. The distribution of meta3C reads along the longest contigs (>100 kb) of the different assemblies confirms that most of the PE reads correspond to 'shotgun'-like reads, with very few useful trans, or long-range cis contacts (*Figure 1—figure supplement 1b*). These 3D ratios compare favorably to those obtained in recently published works, including using commercial solutions that encompass a biotin enrichment step (compare with 0.36% and 2.38% from *DeMaere et al., 2020*; *Press et al., 2017*, respectively), but they remain weak overall.

To increase the 3D ratio, we tested derivatives of metagenomic Hi-C protocols (I and II), corresponding to the meta3C protocol but with the addition of a biotin enrichment step ('Materials and methods'; see also *Cockram et al., 2021*). We applied them on samples #16016 (protocols I and II), #8015 (protocol II), and #9010 (protocol II) (*Figure 1a*). Simply adding a biotin enrichment step to meta3C (protocol I) only marginally improved the ratio (14.9% and 5.9% compared to 4.4% and

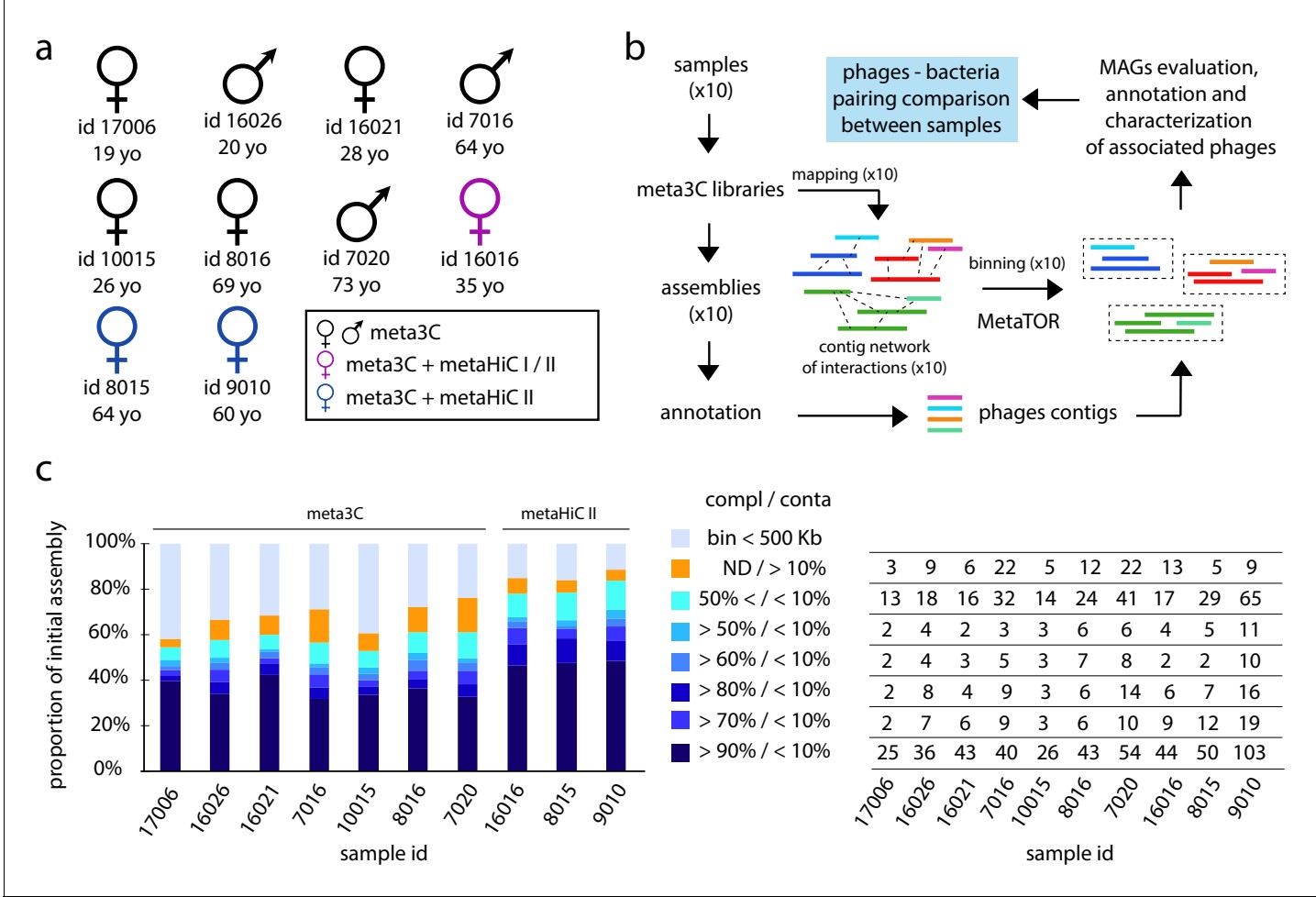

**Figure 1.** Binning of human gut microbiota using 3C and HiC protocols I and II. (**a**) Sex, ID, and age of the 10 individuals investigated in this study. The colors indicate the proximity ligation protocol used to generate the libraries. (**b**) Computational pipeline of paired-end (PE) sequence analysis (MAG: metagenomic assembled genome). (**c**) Left diagram: proportion for each individual assembly of the different bins obtained. Colors represent groups of bins of similar quality (completion/contamination), according to the color code on the right. ND = not determined. Right table: for each individual number of MAGs corresponding to the left diagram.

The online version of this article includes the following figure supplement(s) for figure 1:

**Figure supplement 1.** Comparisons of 3C and Hi-C protocols I and II.
**Figure supplement 2.** Relation between abundance and completion/contamination for the retrieved metagenomic assembled genomes (MAGs).
**Figure supplement 3.** Comparisons of metagenomic assembled genomes (MAGs) obtained using meta3C, metaHiC protocol I, or metaHiC protocol II.

5.1% for the HpaII and MluCI libraries, respectively; *Figure 1—figure supplement 1b*). However, protocol II applied on samples #16016, #8015, and #9010 (using a ligation at room temperature) yields an average 3D ratio of ~32% (with a min/max of 25% and 41%, respectively). This is a strong improvement compared to the meta3C ratio ranging from the same sample (mean = 8.4%; min/max of 4.4% and 12.2%, respectively).

Therefore, protocol II yielded approximately four times more informative trans-contacts between pairs of contigs compared to meta3C, for a same cost. These experiments show that the ligation step was one of the main limiting steps in the Hi-C protocol applied to human stool samples and should be considered carefully in future experiments. The resulting protocol was named metaHiC.

## Binning of human gut microbiome using meta3C or metaHiC protocols

We applied metaTOR on the assemblies and the corresponding contact data generated by either meta3C or the metaHiC protocols I and II. MetaTOR is an iterative network segmentation pipeline

based on the Louvain algorithm (*Blondel et al., 2008*) to identify communities of contigs (or bins) enriched in genuine trans-contacts and likely to correspond to individual genomes for each sample (*Figure 1b* and 'Materials and methods'; *Baudry et al., 2019*; *Marbouty et al., 2014*). Among the 1,485,156 contigs (3.45 Gb) present in the 10 assemblies, 300,677 (525 Mb) were grouped by meta-TOR into 19,122 small bins ranging in size between 10 and 500 kb. An additional 855,633 (2.57 Gb) contigs were grouped into 1,100 larger bins containing more than 500 kb and further referred to as metagenomic assembled genome (MAG) (*Supplementary files 4* and *5*). For each sample, the corresponding MAGs represent between 72% and 91% of all mapped reads. The composition of MAGs was assessed using CheckM, which quantifies the contamination and completion of pools of contigs with respect to known bacterial genomes (*Parks et al., 2015*). Out of the 1,100 MAGs, 304 were identified by CheckM as 'high quality', that is, displaying a completion rate ≥90% and contamination rate ≤5%, and corresponding most likely to complete bacteria genomes, and 411 as "medium quality" (completion ≥50% and contamination ≤10%) (*Figure 1c*). Some of these 715 high- and medium-quality genomes belong to species with an abundance below ~0.1% (over the 10 individuals) (*Figure 1—figure supplement 2a-b*), paving the way to future in-depth individual sample analysis.

We focused on the contribution of the improved metaHiC protocol II to the generation of high-quality MAGs. Indeed, a high 3D ratio does not necessarily correspond to an enrichment in informative bridges between contigs, as high levels of noise (i.e. ligation events between random restriction fragments) could also increase the ratio. However, all metrics (number of high-quality MAGs, completion, etc.) recovered when comparing the binned metagenomic maps generated using metaHiC protocol II with their meta3C or protocol I counterparts point at a strong enrichment in informative contacts (*Figure 1—figure supplement 1c*; 'Materials and methods'). These results are further supported by the modularity for the corresponding networks (0.89, 0.79, and 0.93 for metaHiC protocol II compared to 0.73, 0.71, and 0.86 with meta3C) (*Newman, 2006*), pointing at a better-defined community structure when using the network generated by metaHiC protocol II. A comparison between the MAGs retrieved using either the meta3C, metaHiC protocol I or II datasets showed that all approaches are highly concordant (*Figure 1—figure supplement 3*). The meta3C protocol is therefore the least enriched in useful 3D signal, but this can be overcome either by a higher sequencing depth or by using the metaHiC protocol II.

Therefore, the metaHiC protocol II presented here is several times cheaper than conventional meta3C or current commercial solutions and should be chosen in future studies. The rest of the study was pursued by merging the libraries generated for each individual.

## Diversity of the human gut microbiota assessed by metaHiC/meta3C

We generated a phylogenetic tree of the 715 high- and medium-quality MAGs (*Figure 2*), as well as for each individual sample (*Figure 2—figure supplement 1*). As expected, the trees cover the taxonomic groups found in a healthy human intestinal tract (*Almeida et al., 2019*). In addition, the abundance of the different MAGs is consistent with the outcome of taxonomic analysis of the raw reads by the classifier Kaiju (*Menzel et al., 2016*; *Figure 2—figure supplement 2*).

The 715 MAGs were then compared to the latest genome references of the GTDB-Tk databases (release 0.95; *Chaumeil et al., 2020*). Eighty-five (around 10%) of them did not present any close similarity to the GTDB-Tk references at a threshold identity of 95% (average nucleotide identity; *Konstantinidis and Tiedje, 2005*). New MAGs are proportionally distributed among the different clades, with a majority belonging to Clostridia, in agreement with *Pasolli et al., 2019*. This relatively low number of newly recovered species, although still significant, suggests that the databases about the human gut microorganisms (at least those describing western population) are converging toward completion.

Note that, starting from 200 million PE reads on a single sample (#9010), the metaHiC protocol II was able to catch 81 high-quality genomes including seven from unknown species. This result therefore demonstrates the interest of this approach when investigating complex microbial communities from single samples.

## Phages-hosts network in the human intestinal tract

We used VIRSorter (*Roux et al., 2015*) and VIBRANT (*Kieft et al., 2020*) to annotate phage sequences among the 1,485,156 contigs generated from the 10 assemblies. VIRSorter and VIBRANT

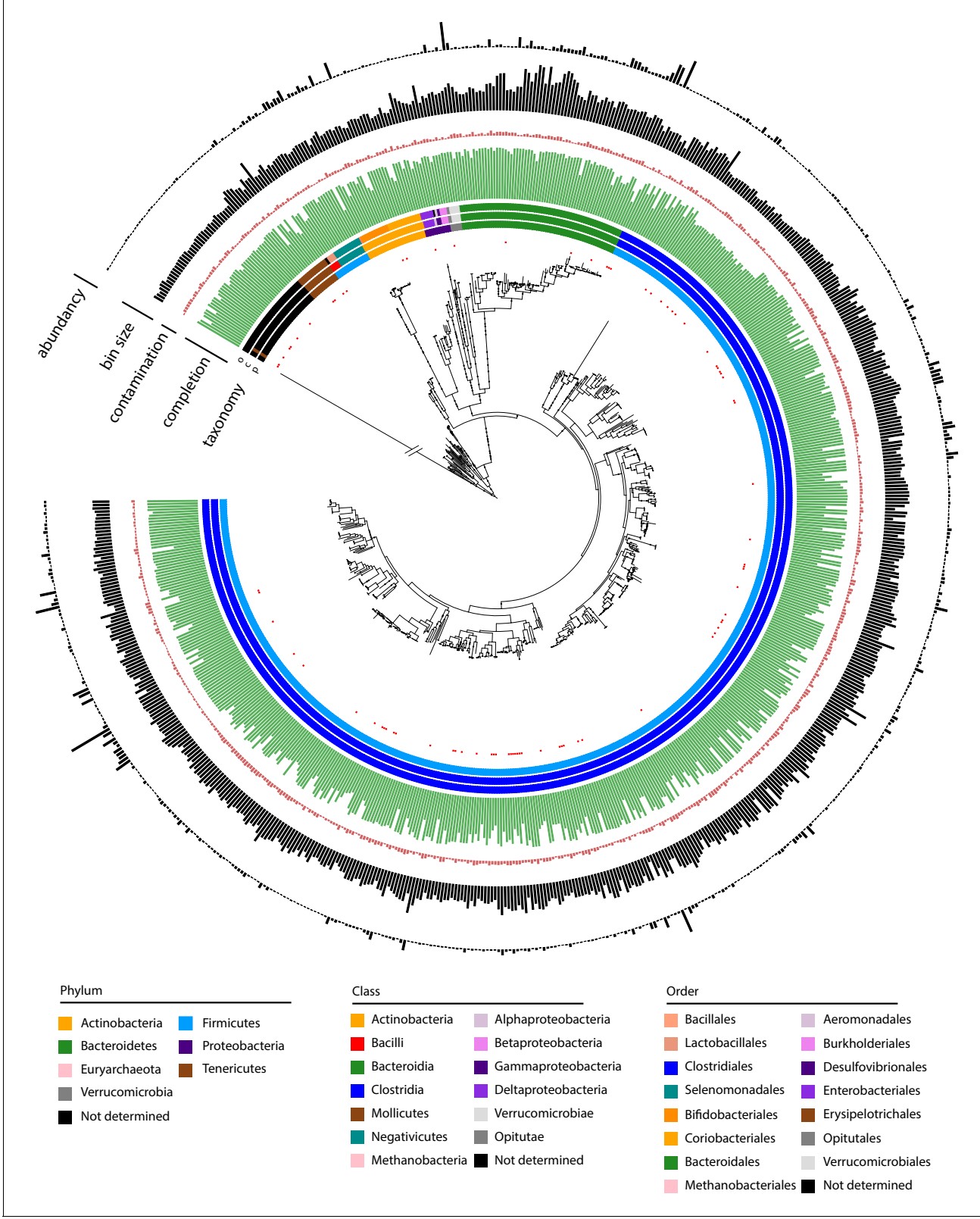

**Figure 2.** Metagenomic assembled genomes (MAGs) recovered using meta3C/metaHiC applied on 10 healthy human gut samples. Phylogenetic tree comprising the 715 reconstructed MAGs with a completion above 50% and a contamination below 10% (691 different species at a threshold of 95% identity). A very long branch was cut out (symbol / /) to resize the tree for clarity. Red dots indicate MAGs with no reference in the GTDB-tk database (threshold of homology = 95%). The colors of the first three layers indicate the taxonomy of the MAGs, as determined by CheckM (phylum, class, order

*Figure 2 continued on next page*

*Figure 2 continued*

from center to periphery). Green and red bars represent completion and contamination levels, respectively. Completion scale: max = 100%; min = 50%. Contamination scale: max = 10%; min = 0%. Gray bars: MAGs (bin) sizes (scale: max = 7.07 Mb, min = 766 kb). Black bars in the outmost layer indicate MAG abundance in the different samples (max = 4.73%; min = 0.0039%).

The online version of this article includes the following figure supplement(s) for figure 2:

**Figure supplement 1.** Phylogenetic tree for the 10 processed samples.

**Figure supplement 2.** Comparisons of metagenomic assembled genomes (MAGs) taxonomic abundance and reads taxonomic abundance.

identified 4,649 and 7,918 putative phages, respectively (including all categories for VIRSorter and excluding identified prophages, see 'Materials and methods'), resulting in 9,488 unique annotated phages contigs, ranging in size between 1 and 10 kb. Taxonomic annotations of those sequences using DemoVir (https://github.com/feargalr/Demovir) confirmed the high prevalence of phages belonging to the Caudovirales order (families Siphoviridae 59%, Myoviridae 15%, Podoviridae 2%, Unassigned 22%). Trans-contacts between phage annotated contigs and the 1,100 MAGs unveiled the infection pattern of the different phages presents in the 10 samples (*Marbouty et al., 2014*; *Marbouty et al., 2017*). Three classes (A, B, C) of phages were defined (*Figure 3—figure supplement 1*; see also 'Materials and methods'). Class A corresponds to 6,503 phages contigs (out of 9,488) that can be unambiguously assigned to a single MAG based on their 3D contacts (*Supplementary file 6*). In total, 955 MAGs (out of 1,100) made specific contacts with one or more of these contigs. Class B corresponds to the 2,460 phage annotated contigs assigned to highly contaminated bins prior to recursive procedure, and class C to the 525 contigs that never cluster with a bin above 500 kb during the 100 iterations of the iterative partition procedure. This latter category corresponds either to free phages or to phage sequences associated with uncharacterized/poorly abundant MAGs. Among the 2,460 contigs from category B, 364 clustered with a MAG at least once over the 10 Louvain recursive partition procedure, with 260 of those displaying an association score of at least 5/10 (same MAG for 5 out of the 10 iterations of the recursive procedure). These 260 contigs were reintroduced into class A for a total number of phage annotated contig in this class of n = 6,763. The 2,200 remaining phages contigs of class B (i.e. potentially contacting multiple MAGs) and the 525 phage contigs from class C (making no contacts at all with large MAGs) may correspond to free phages. These results suggest that the majority (6,763 out of 9,488, i.e. ~70%) of the phages present within the human gut are specific, infecting only one bacterial species (*Figure 3a*).

On average, a MAG was associated with 6.9 phages contigs, but this rate presented high fluctuations (SD = 7.4), with a maximum of 45 contigs (MAG #9010_74_0; *Supplementary file 4* and *6*). No correlation between this rate and taxonomic annotation was identified.

We tested the coherence of the host-phage pairing strategy by focusing on phylogenetically phages related (i.e. homologous) sequences to see whether phages-MAGs pairs were conserved across individual samples. We searched for pairs of class A phages contigs belonging to the same genus, but with each member of the pair belonging to a different sample out of the 10 individuals (i.e. sharing at least 70% identity on 70% of their length; *Lavigne et al., 2008*). Among 454 pairs identified, 394 of them corresponded to phages assigned to two different MAGs (from the corresponding individual) but presenting the same taxonomic assignment (*Figure 3—figure supplement 2a*). In addition, three phylogenetically related MAGs from three different individuals exhibited the same set of homologous phages assigned to them (*Figure 3—figure supplement 2b*). These results support a good level of confidence in the phage-host network.

The network was further corroborated by looking for CRISPR spacer matches between MAGs and phages contigs using PILER-CR and Blast over the entire network (all individuals data mixed; *Edgar, 2007*; *Edwards et al., 2016*). We found 1,252 robust spacer signatures shared by a MAG and one or more phages sequences ('Materials and methods'). Of those, 387 were associated to a host-phage pair from the same individual. Among the 865 remaining hits, 90% (781/865) corresponded to MAGs and phages contigs from different individuals; 84% of these hits (660/781), however, involved MAGs from two different individuals but with the same taxonomy at the genus level compared to the characterized hosts in the same subject. This suggests that these MAGs have already been in contact with these phages during their evolutionary history but are not in the present individual.

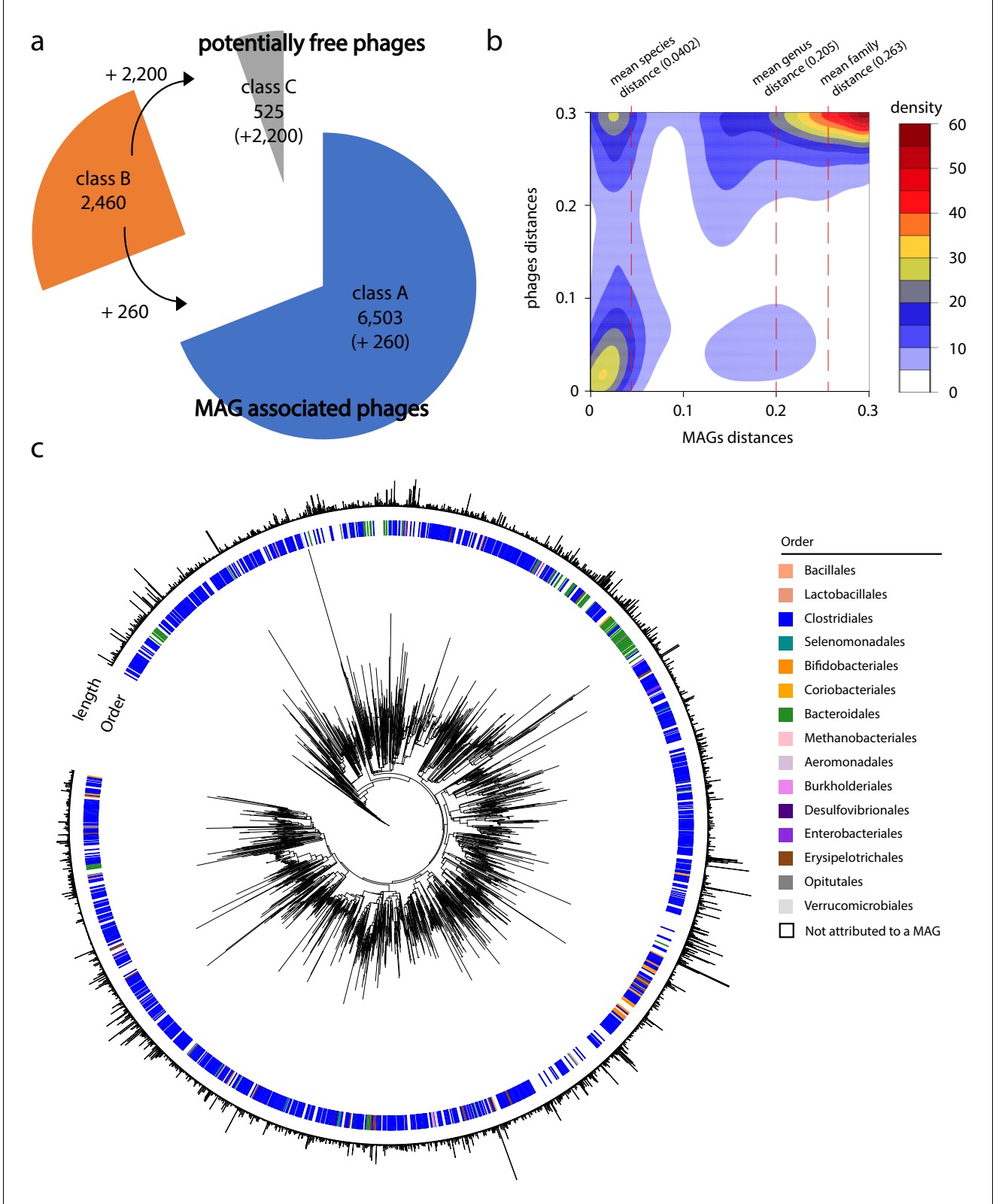

**Figure 3.** Phage-bacteria network of interactions in human gut. (a) Pie chart of phages contigs distribution among the different classes (see 'Materials and methods' and *Figure 3—figure supplement 1*). (b) Gaussian smoothed heatmap of the sequence distance between all phages taken two by two (Y-axis) and the sequence distance between their associated metagenomic assembled genomes (MAGs) (X-axis). Distance varies between 0 (full sequence similarity) and 0.3 (no similarity). Color scale on the left indicates the counting. Dotted lines indicate mean value of distances between

*Figure 3 continued on next page*

*Figure 3 continued*

MAGs at different taxonomic levels (species = 0.0402, genus = 0.205, family = 0.263). (**c**) Phylogenetic tree of phage contigs (n = 2,726). Colored strips in the first layer indicate the taxonomy of the associated MAG at the order level (1,535 over the 2,726 contigs present in the tree are attributed to a MAG). Bar plot in the second layer indicates phages contigs size (min = 5 kb; max = 211 kb).

The online version of this article includes the following figure supplement(s) for figure 3:

**Figure supplement 1.** Classification of phages contigs.

**Figure supplement 2.** Comparison of phages and their characterized hosts between samples.

To further investigate the infection network, we computed the sequence distance between all MAGs and phages contigs using the program Mash (*Ondov et al., 2016*). We then generated a density plot of the distance between all phages sequences as a function of the distance between the sequences of their related MAGs (*Figure 3b*). The plot showed that homolog phage sequences are associated to homolog MAGs sequences (*Figure 3b*, bottom left corner). In addition, while distantly related phages are occasionally found infecting homologous MAGs (*Figure 3b*, top left corner), very few related phages infect non-homologous MAGs (*Figure 3b*, bottom right corner). This result supports the idea that phages are 'locally adapted' to their bacterial hosts and do not easily shift from one bacterial species to another (*Koskella and Meaden, 2013*; *Vos et al., 2009*). Plotting the mean distance score between MAGs sequence for different taxonomic levels (species, genus, families; *Figure 3b* – dashed vertical lines) highlights a disruption at the species level, further supporting the presence of interspecific boundaries.

Phage contigs were then positioned in a phylogenetic tree based on a set of 77 specific markers of double-stranded DNA phages (*Low et al., 2019*) ('Materials and methods'). The genus taxonomic level of the host, when identified, was then indicated on the tree (*Figure 3c*); 2,726 out of the 9,488 phages contigs, including 1,535 with an assigned host, contained at least three markers and were included in the tree. The tree unveils clusters of closely related phages infecting the same bacterial genus (Kruskal-Wallis test, p-value=0.0132) suggesting, again, that phages are specific to their hosts in human gut.

## Phages-host ratio in the human intestinal tract

The phage-bacteria network has the potential to provide insights on the phages' metabolism. For instance, the proportion of infected cells in the host population is unknown; but, comparing the read coverages of a phage contig and its associated MAG can provide indication about the former's activity (*Figure 4*). Indeed, the genome of an actively replicating phage should, in theory, display a higher coverage than that of its bacterial host (although multiple factors will nevertheless influence the phage sequence coverage within a metagenome, including the number of copies or sequence composition bias). For the 6,763 class A phages contigs, their mean read coverage ratio compared to the MAGs ratio is 1.23 (SD = 1.70), with a maximum of 58. This distribution is similar to the one of all the contigs binned within our different MAGs (*Figure 4A,a*). There are nonetheless significant variations among the 10 individual microbiota, as the average ratio varies from 1 (sample #10015) to 2 (sample #17006) (*Figure 4—figure supplement 1a*). These variations could reflect differences between individual gut microbiomes or through time, which should be elucidated using larger cohort and time-series sampling. We further investigated the potential influence of GC% of the sequence on the ratio. Plotting the coverage ratio of the phages contigs as a function of the mean GC% of these contigs, or as a function of the GC ratio (phages contigs/MAGs mean GC%), did not unveil a correlation between the ratio and GC% (*Figure 4—figure supplement 1b*).

Then, we investigated the influence of replication activity in the host by searching in each associated MAGs contigs encompassing the *dnaA* gene, and therefore likely to carry the origin of replication of the bacterial genome (n = 856 MAGs implicating 6,239 phages contigs) (*Emiola and Oh, 2018*). As expected, 'dnaA' contig (dubbed oriMAG contig) exhibits a mean read coverage ratio higher than the other contigs (mean = 1.37; SD = 0.51; max = 5.15). The coverage ratio analysis was then refined considering the phage vs. oriMAG contigs coverage ratio instead of the mean coverage of the MAGs (*Figure 4Aa*). We classified the phages into four categories based on their read coverage ratio with oriMAGs contigs (*Figure 4a and b*).

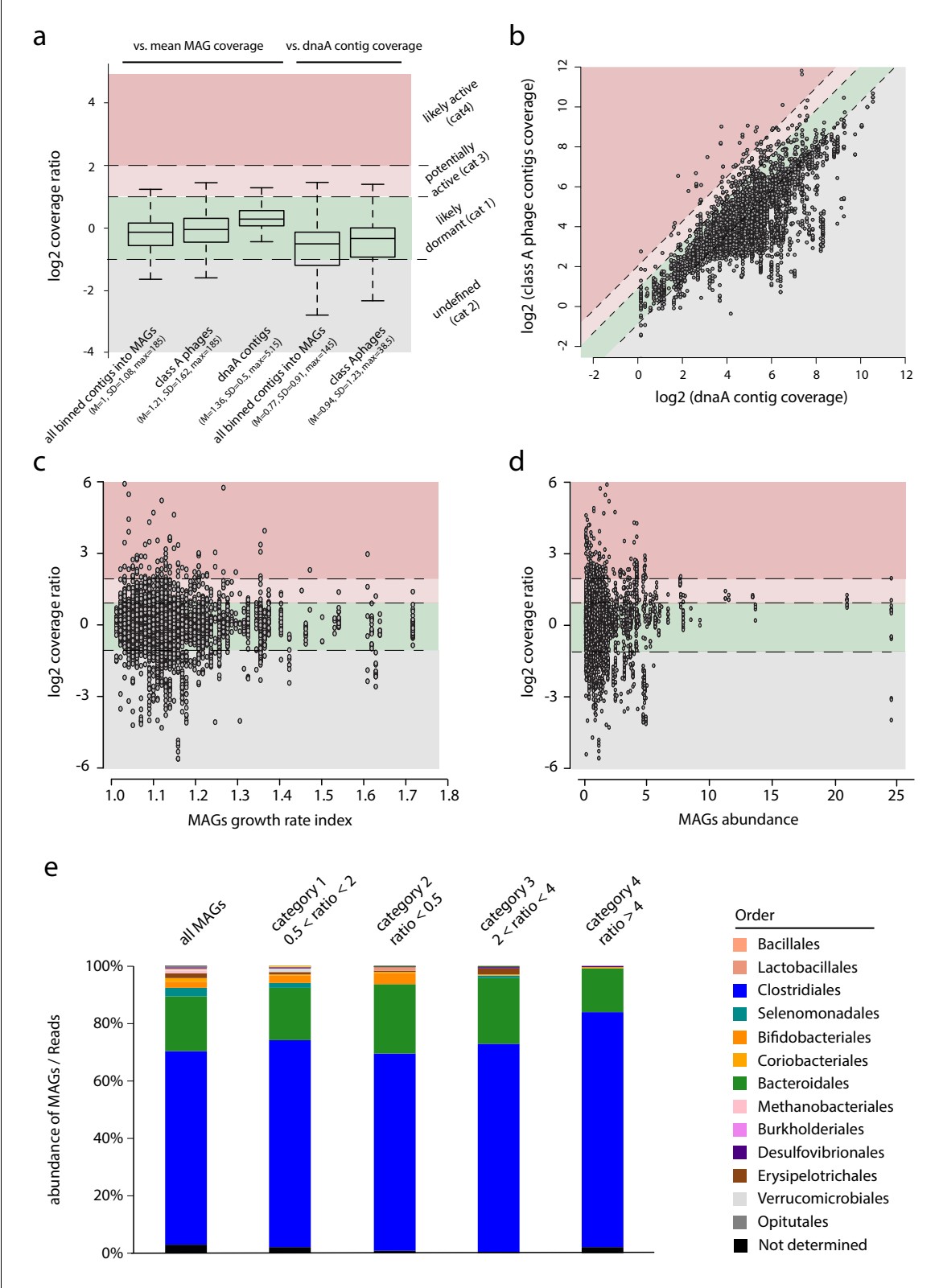

**Figure 4.** Phages-host ratio in human intestinal tract. (**a**) Boxplot of the coverage ratio between different classes of contigs and the mean coverage of their associated MAGs. From left to right, coverage ratio of: (1) all binned contigs into metagenomic assembled genomes (MAGs) (n = 861,082) vs. mean MAG, (2) class A phage contigs (n = 6,763) vs. mean MAG, (3) dnaA contigs (n = 856) vs. mean MAG coverage, (4) all binned contigs into MAGs (n = 672,156) vs. dnaA contig of their associated MAGs, (5) class A phage contigs (n = 6,239) vs. dnaA contig of their associated MAGs. Log2 ratio

*Figure 4 continued on next page*

*Figure 4 continued*

thresholds define colored areas reflecting the potential phage status: green, dormant (cat1: −1 < log2(ratio) <1); gray, undefined (cat2: log2(ratio) < −1); light red, potentially active (cat3: 1 < log2(ratio) <2); red, likely active (cat4: log2(ratio) >2). Inside bar of the box, median; box, quartiles; whiskers, 1.5× interquartile range. (b) Read coverage of phage contigs (n = 6,239) as a function of the read coverage of the dnaA contig of their associated MAG (n = 856). Dashed lines indicate the limit of the categories described in a. (c) Scatterplot of class A phage contigs coverage vs. dnaA contig coverage as a function of the growth rate index calculated using GRID for each MAG. (d) Scatterplot of class A phage contigs coverage vs. dnaA contig coverage as a function of the MAGs abundance. (e). Bar plot of MAGs taxonomic abundance depending on the categories specified in (a).

The online version of this article includes the following figure supplement(s) for figure 4:

**Figure supplement 1.** Phages-hosts ratio.

Phages displaying a ~1:1 ratio with their oriMAG contig (category 1, 0.5 < ratio < 2.0) represented the vast majority of the 6,239 phages-host pairs (n = 4,477; ~75%). They correspond to likely dormant (pro)phages, able to reactivate or not. This population may also encompass active phages that incidentally exhibit the same coverage of their host. Then, for 1,395 phage-hosts pairs, the corresponding oriMAG contig was significantly more covered than the associated phage (category 2 – undefined). This profile is consistent with an abortive phage infection cycle, a pseudo-lysogenic cycle, or even with prophages not present in the whole host population. The remaining 367 pairs (~5%) corresponded to phages with a higher coverage than their assigned oriMAG contig: category 3 corresponded to a ratio between 2 and 4 (n = 291) and category 4 to a ratio ≥4 (n = 76). These phages may be (category 3) or are probably (category 4) actively replicating, potentially impacting the ecosystem through lytic activities. Overall, these results support the conclusions of former and recent work suggesting that the major part of the phages found in human gut are temperate phages with few lytic activity (*Džunková et al., 2019*; *Minot et al., 2011*; *Reyes et al., 2010*).

To further question the influence of active phages, we investigated MAGs abundance and growth rate using GRID. This program measures for each MAG the read coverage ratio between the contigs (if any) that carry either the *dnaA* gene (i.e. oriMAG contig) or the marker of replication termination site *dif* (*Emiola and Oh, 2018*). No significant correlation was detected between the presence of active phages and the abundance/growth rate of their corresponding hosts, as assessed by this analysis (*Figure 4c and d*). We then asked whether some bacterial genera are more or less prone to phages activity (*Figure 4e*). Bifidobacteriales and Selenomodales appear underrepresented in the targeted MAGs, which suggests that they could be less subject to phages lytic activity in the gut than other genera.

## CrAss-like phages family in the human intestinal tract

CrAss-like phages represent a large family of phages widespread in the human gut. They are predicted to infect bacteria from the Bacteroides phylum (*Dutilh et al., 2014*; *Guerin et al., 2018*; *Shkoporov et al., 2018*; *Yutin et al., 2018*). So far, only two hosts have been described 'in vitro' for one crAss member: *Bacteroides intestinalis* (*Shkoporov et al., 2018*) and *Bacteroides thetaiotaomicron* (*Hryckowian et al., 2020*). We searched for the presence of crAss-like phages in the different individual assemblies. Using a set of representative crAss-like phages sequences (*Guerin et al., 2018*), we detected 17 contigs, ranging in size from 58 to 176 kb and presenting several proteins homologs to the crAss-like family ('Materials and methods'). The phylogenetic tree encompassing these 17 contigs and the set of representative sequences pointed at phages belonging to genera I, III, IV, VI, and V (*Figure 5* and *Supplementary file 3*; *Low et al., 2019*). All these contigs were unambiguously assigned to MAGs belonging to the Bacteroides clade, as expected from previous predictions and co-culturing (*Dutilh et al., 2014*; *Guerin et al., 2018*; *Hryckowian et al., 2020*; *Shkoporov et al., 2018*). More specifically, *Bacteroides vulgatus*, *Bacteroides uniformis*, *Bacteroides ovatus*, and *Bacteroides thetaiotamicron* were identified as hosts of these crAss-like phages. These four species were already suspected to be, or characterized as, hosts for crAss-like phages (*Guerin et al., 2018*; *Hryckowian et al., 2020*; *Shkoporov et al., 2018*). Interestingly, we also detected unknown species from the Prevotellaceae and Porphyromonadaceae families as hosts of other crAss-like phages. Some of these phages come from the same individual and were assigned to the same MAG, suggesting that crAss-like phages are highly abundant and potentially competing for hosts in the human gut.

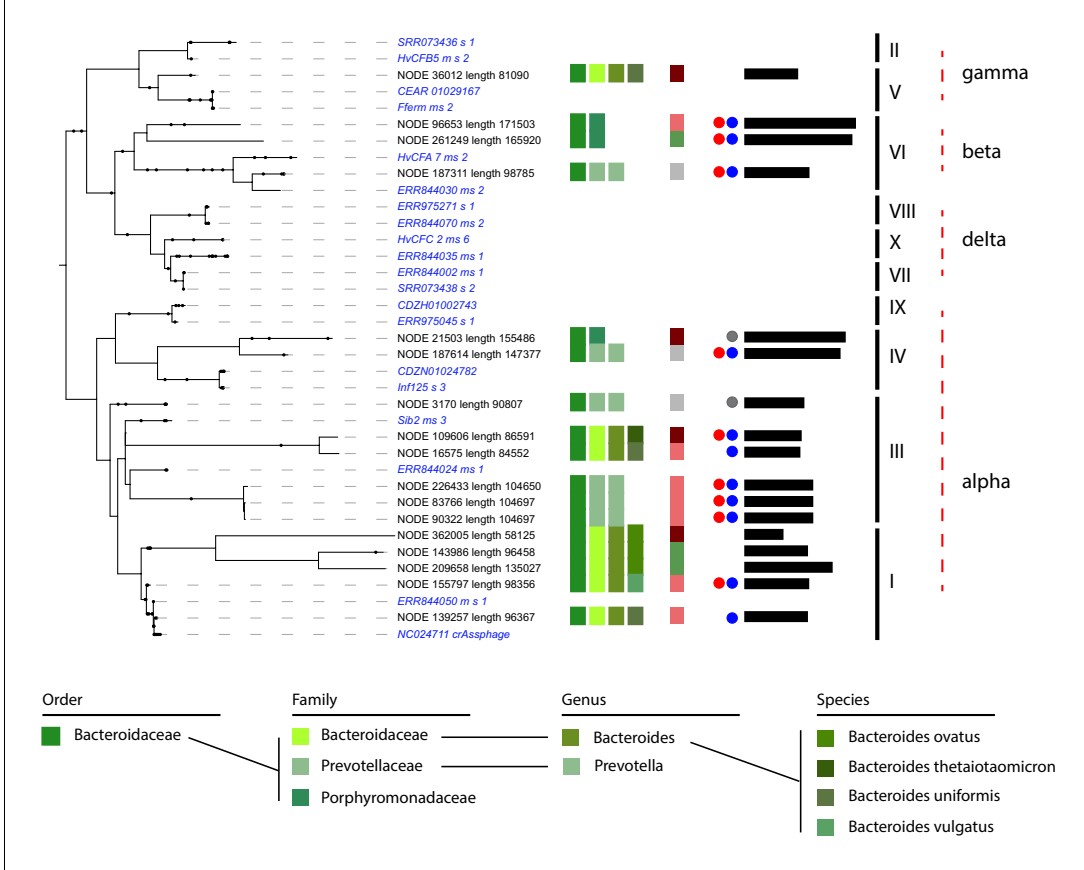

**Figure 5.** CrAss-like phages and their associated hosts. Phylogenetic tree of crAss-like phages contigs found in the 10 assemblies. Representatives of the 10 genera of crAss-like phages described by Guerin et al. are included (in *italic blue*). Names of the different contigs are indicated on the left of each branch. Vertical colored stripes indicate, from left to right: host metagenomic assembled genome (MAG) order, host MAG family, host MAG genus, host MAG species, ratio of the read coverage of phages vs. read coverage of MAG (red – ratio > 4; light red – 4 > ratio > 2; green – 2 > ratio > 0.5; gray – ratio < 0.5). Red dot: circular signal characterized by VirSorter. Blue and gray dot: strong and weak circular signal characterized by 3C, respectively. Black lines: size of the contigs (min = 58,125,125 pb; max = 171,503 bp). Genera of the crAss-like phages are indicated on the right of the tree.

The online version of this article includes the following figure supplement(s) for figure 5:

**Figure supplement 1.** CrAss-like phages contact maps.

Among these 17 contigs, nine were predicted to be circular by VirSorter. Indeed, a circular signal was visible in their 3D contact map (*Figure 5—figure supplement 1*). 3D contacts also pointed at four more circular crAss-like phage contigs. Finally, the phage-host ratio of these 17 contigs showed that crAss-like phages are an active family of phages, with 11 contigs exhibiting a ratio ≥2 and therefore likely multiplying. On the other hand, we could not determine whether these ratios were associated to a low growth rate, and further analysis will be needed to understand the impact of crAss-like phages on their hosts. Altogether, these results significantly extend our understanding of the spreading of the ubiquitous but elusive crAss-like family, while confirming its link with Bacteroides.

## Discussion

Bacteriophages, also called bacteria viruses, are the most abundant biological entities on earth and have a major impact on microbial communities (*Clokie et al., 2011*). High-throughput DNA sequencing of environmental samples has allowed to bypass virus isolation and revealed the existence of a

wide variety of phages in all ecosystems (*Cornelissen et al., 2012*; *Devoto et al., 2019*; *Dutilh et al., 2014*). Several studies and computational developments have shed light on the composition and dynamics of the viral part of the human gut (*Manrique et al., 2016*; *Minot et al., 2013*; *Shkoporov et al., 2019*), unveiling entirely new phages family such as the crAss-like family (*Dutilh et al., 2014*; *Guerin et al., 2018*; *Yutin et al., 2018*). However, the concomitant investigation of the relationships between phages and bacteria remains challenging (*Džunková et al., 2019*; *Marbouty et al., 2017*). Moreover, metagenomic approaches often rely on covariance analysis involving multiple sampling, which can be cumbersome. Proximity ligation assays applied to metagenomic samples (meta3C, metaHiC) allow in situ investigation of microbial communities starting from single samples (*Marbouty et al., 2014*; *Marbouty et al., 2017*). In the present study, we applied those approaches to 10 human gut microbiota samples, characterizing hundreds of high-quality bacterial genomes (MAGs) as well as thousands of phage sequences and revealing the large multiscale network of interactions between them.

The approach unveiled hundreds of MAGs, with ~10% of them identified as new, a relatively low number. However, given the in-depth investigation of the gut with metagenomic studies, this is not surprising. It is a dynamic field: in an earlier version of this work, nearly 30% of the MAGs identified in the present study were considered as 'new'. The exhaustive analysis of *Pasolli et al., 2019* reduced that number to 10%. That number remains also significant (85 MAGs), given only 10 individuals were screened and no covariance approach performed. Therefore, there is still in our opinion a strong potential in the approach to quickly call new, complete MAGs de novo from complex microbial ecosystems not as intensively sampled as the human gut. This prediction is likely to hold true until long-read sequencing, which provides complete or near complete genomes at a high frequency, becomes the prevalent choice of future metagenomic studies. The ability to assign episomes to their hosts will nevertheless remain an asset of metaHiC in the future.

The phage-host ratio computed in this study suggests that phages appear slightly more abundant than bacteria in the human intestinal tract but nothing comparable to other environments (*Rodriguez-Valera et al., 2009*). Therefore, this work confirms that the majority of phages present in the gut are temperate phages with little or no lytic activity (*Reyes et al., 2010*). This result is in adequation with previous studies that showed a limited phage-host dynamic and a relative homeostasis of this environment, with few examples of a phage overrepresented compared to its host. On the other hand, nearly 5% of the phage population exhibits a sequence coverage ratio higher than twice that of their hosts, representing actively replicating phages that could play an important role in the maintenance of the bacterial population. Notably, members from the Bifidobacteriales and the Selenomonadales bacteria clades appear to be less subject to phages lytic activity. The work also confirms that most of the phages are specific to their host and do not shift easily to another host, at least in the human intestinal tract (*Koskella and Meaden, 2013*; *Vos et al., 2009*).

As expected from their overrepresentation in the human gut population (*Edwards et al., 2019*), we found several potential members of the crAss-like phages family. Those phages and their hosts have been sought in vitro since their discovery in 2014, with two successful attempts (*Hryckowian et al., 2020*; *Shkoporov et al., 2018*). In a single analysis, meta3C/HiC unveiled the host, at the species level, of 17 crAss-like contigs, all belonging to different members of the Bacteroides clade. This result is in agreement with previous prediction at the class level (*Dutilh et al., 2014*), and suggests crAss-like phages are widely distributed in the Bacteroides clade.

It also underlines the interest of this affordable approach, which is easily applicable to human gut samples. Application of metaHiC on larger cohorts, including or not longitudinal tracking, will allow to better understand the relationships not only between crAss-like phages and this clade but also the specificity of each phages sub-group to their hosts in the Bacteroides clade. CrAss-like phages share features characteristics to both phages and plasmids and generally encode proteins allowing for their own replication. The crAss-like phages are therefore reminiscent of the concept of plasmid or phage plasmid (*Lagos and Goldstein, 1987*).

Finally, coupling metaHiC with other approaches on larger human cohort will shed new lights and findings about the relationships between phages, plasmids, and bacteria. For instance, coupling meta3C with VLP purification could offer an unprecedent view of phage-bacteria relationships in this important ecosystem.

# Materials and methods

## Key resources table

| Reagent type (species) or resource | Designation | Source or reference | Identifiers | Additional information |
|---|---|---|---|---|
| Biological sample (human) | Frozen human fecal samples | Institut Pasteur Biobanque (ICAReB) | Patient 17006–19 yo – female | Freshly frozen from healthy donor |
| Biological sample (human) | Frozen human fecal samples | Institut Pasteur Biobanque (ICAReB) | Patient 16026–20 yo – male | Freshly frozen from healthy donor |
| Biological sample (human) | Frozen human fecal samples | Institut Pasteur Biobanque (ICAReB) | Patient 16021–28 yo – female | Freshly frozen from healthy donor |
| Biological sample (human) | Frozen human fecal samples | Institut Pasteur Biobanque (ICAReB) | Patient 7016–64 yo – male | Freshly frozen from healthy donor |
| Biological sample (human) | Frozen human fecal samples | Institut Pasteur Biobanque (ICAReB) | Patient 10015–26 yo – female | Freshly frozen from healthy donor |
| Biological sample (human) | Frozen human fecal samples | Institut Pasteur Biobanque (ICAReB) | Patient 8016–69 female | Freshly frozen from healthy donor |
| Biological sample (human) | Frozen human fecal samples | Institut Pasteur Biobanque (ICAReB) | Patient 7020–73 yo – male | Freshly frozen from healthy donor |
| Biological sample (human) | Frozen human fecal samples | Institut Pasteur Biobanque (ICAReB) | Patient 16016–35 yo – female | Freshly frozen from healthy donor |
| Biological sample (human) | Frozen human fecal samples | Institut Pasteur Biobanque (ICAReB) | Patient 8015–64 yo – female | Freshly frozen from healthy donor |
| Biological sample (human) | Frozen human fecal samples | Institut Pasteur Biobanque (ICAReB) | Patient 9010–60 yo - female | Freshly frozen from healthy donor |
| Chemical compound, drug | Formaldehyde (35–37%) | Sigma Aldrich | F8775 | (also contain methanol as stabilizer – 15%) |
| Chemical compound, drug | MyOne streptavidin beads | Life Science | | |
| Chemical compound, drug | dCTP-14 Biotin | Life Science | | |
| Software, algorithm | Cutadapt | *Martin, 2011* | v.1.9.1 | |
| Software, algorithm | FastQC | *Andrews, 2010* | v.0.10.1 | |
| Software, algorithm | MEGAHIT | *Li et al., 2015* | v.1.1.1.2 | |
| Software, algorithm | Quast | *Gurevich et al., 2013* | v.2.2 | |
| Software, algorithm | bowtie2 | *Langmead and Salzberg, 2012* | v.2.2.3 | |
| Software, algorithm | MetaTOR Pipeline | *Baudry et al., 2019* | | |
| Software, algorithm | MetaBat | *Kang et al., 2015* | | |
| Software, algorithm | CheckM | *Parks et al., 2015* | v1.1.2 | |
| Software, algorithm | GTDB-Tk | *Chaumeil et al., 2020* | release 0.95 | |
| Software, algorithm | seqtk | *seqt, 2020*; https://github.com/lh3/seqtk | | |
| Software, algorithm | hicstuff | *Matthey-Doret, 2020*; https://github.com/koszullab/hicstuff | | |

*Continued on next page*

*Continued*

| Reagent type (species) or resource | Designation | Source or reference | Identifiers | Additional information |
|---|---|---|---|---|
| Software, algorithm | LAST | http://last.cbrc.jp/ | | |
| Software, algorithm | itol | https://itol.embl.de/ | | |
| Software, algorithm | VirSorter | *Roux et al., 2015* | v.1.0.3 | |
| Software, algorithm | VIBRANT | *Kieft et al., 2020* | v.1.0.1 | |
| Software, algorithm | Mash | *Ondov et al., 2016* | v.2.0 | |
| Software, algorithm | PILER-CR | *Edgar, 2007* | | |
| Software, algorithm | Prodigal | *Hyatt et al., 2010* | v.2.6.3 | |
| Software, algorithm | MUSCLE | *Edgar, 2004* | | |
| Software, algorithm | AMAS | *Borowiec, 2016* | | |
| Software, algorithm | IQ-TREE | *Nguyen et al., 2015* | v.1.5.5 | |
| Software, algorithm | R environment | *R Development Core Team, 2020* | | |
| Other | Covaris S220 | Covaris | AFA tubes | |
| Other | Precellys TUBE | Bertin Technology | VK05 + VK01 glass beads | |

## Feces sampling and meta3C libraries generation

Frozen feces samples of 10 healthy human adults (19–73 years old) were collected from the Institut Pasteur Bobanque (ICAReB) (agreement N18). For each sample, 100 mg was thawed directly in 50 mL of crosslinking solution (1× PBS supplemented with 5% formaldehyde) and incubated for 1 hr at room temperature under strong agitation. Formaldehyde was quenched by adding 20 mL of 2.5 M glycine during 20 min at room temperature under gentle agitation. Samples were then recovered by centrifugation, washed with 10 mL 1× PBS, re-centrifuged, and aliquots of 20 mg were stored at −80°C until processing. One aliquot (20 mg) of fecal matter was resuspended in 4 mL TE 1× supplemented with antiprotease (mini tablets – Roche), transferred in two Precelys tubes (2 mL – VK05 supplemented with 100 µL of VK01 glass beads) and disrupted (6,700 rpm – 20 s ON/30 s OFF – six cycles). Lysates were recovered, pooled, and 10 µL of Ready-to-lyze lysozyme (Epitech) was added prior a 1 hr incubation at 37°C under gentle agitation. Sodium dodecyl sulfate (SDS) 10% was added to a final concentration of 0.5% and the lysate was incubated 10 min at room temperature. For each library, 1 mL of lysate was transferred to a tube containing the digestion reaction solution (500 µL NEB1 10× buffer, 500 µL Triton 10%, 1000 U HpaII or MluCI, $H_2O$, final volume = 4 mL). Digestion was allowed to proceed for 3 hr at 37°C under gentle agitation. Tubes were then centrifuged for 20 min at 4°C and 16,000 × g and the supernatants discarded. Pellets were resuspended in 500 µL $H_2O$ and transferred to a tube containing the ligation reaction (160 µL NEB ligation buffer 10×, 16 µL ATP 100 mM, 16 µL BSA 10 mg/mL, 500 U T4 DNA ligase, final volume = 1.1 mL). Ligations were processed for 4 hr at 16°C; 20 µL EDTA 0.5M, 80 µL SDS 10%, and 2 mg proteinase K were added to each reaction and incubated overnight at 65°C to digest proteins. DNA was extracted using phenol-chloroform and precipitated with 2.5 vol ethanol 100%. Pellets were suspended in a final volume of 130 µL TE 1× supplemented with RNAse, incubated 1 hr at 37°C and stored at −20°C until use. Libraries were then processed for sequencing as described (*Baudry et al., 2019*; *Marbouty et al., 2017*).

## MetaHiC libraries generation

MetaHiC libraries were generated similarly to 3°C with minor changes. After digestion, tubes were centrifuged for 20 min at 4°C and 16,000 × g, supernatants were discarded, and pellets were resuspended in 400 μL $H_2O$. Biotinylation was done by adding 50 μL NEB ligation buffer 10× (without ATP), 4.5 μL of 10 mM dATP/dTTP/dGTP (HpaII), 37.5 μL biotin-dCTP 0.4 mM, and 8 μL Klenow (5 U/μL) (for MluCI: 10 mM dCTP/dTTP/dGTP and biotin-dATP). Reactions were incubated for 45 min at 37°C. Ligations were performed at 16°C for 4 hr in the case of the metaHiC protocol I and at room temperature for 3 hr in the case of the metaHiC protocol II. DNA was extracted, purified, and processed into sequencing library as described above (*Moreau et al., 2018*).

## Sequencing, processing, and assembly

Proximity ligation libraries were sequenced using PE Illumina sequencing (2 × 75 bp, NextSeq500 apparatus) (*Supplementary file 1*; *Baudry et al., 2019*). Reads were filtered and trimmed using cutadapt (*Martin, 2011*) (parameters: `-a file:adapters.fa -A file:adapters_illumina.fa -match-read-wildcards-q` *20,20 m 45*). Quality was controlled with FastQC (*Andrews, 2010*; http://www.bioinformatics.babraham.ac.uk). 1,478,113,712 PE reads were retained in total (from the 20 3C libraries) (*Supplementary file 1*). These reads were used to perform for each sample an assembly using MEGAHIT v.1.1.1.2 (*Li et al., 2015*) and default parameters. Contigs under 500 bp were discarded from further analyses. Assemblies were analyzed using Quast (*Gurevich et al., 2013*; *Supplementary file 1*).

## Alignment step and 3D ratio calculation

PE reads were aligned separately in single-end mode using bowtie2 (option: `-very-sensitive-local`) (*Langmead and Salzberg, 2012*). For each sample, both alignment files (forward and reverse) were then sorted and merged using the samtools and pysam libraries. Alignments with mapping quality below 20 and ambiguous alignment were discarded. 3D ratio was then calculated by dividing the numbers of PE reads where each end mapped on different contigs by the total number of mapped PE reads (*Supplementary file 2*).

## MetaTOR binning procedure

For the binning procedure, we applied the metaTOR pipeline (https://github.com/koszullab/metaTOR) (*Baudry et al., 2019*). Briefly, each PE reads that aligned unambiguously and with a mapping score >20 on two different contigs were kept to generate the network of contigs' interactions. Interaction scores between contigs were normalized by dividing their interaction score by the geometric mean of the contigs' respective coverage. Contig coverage was calculated using MetaBat script: jgi_summarize_bam_contig_depths with default parameters for every set of reads and a minimum contig size of 500 bp (*Kang et al., 2015*). We then applied 100 iterations of the Louvain procedure (*Blondel et al., 2008*) and recovered bins that clustered together at least 90 times over the 100 iterations. Bins above 500 kb were evaluated for completeness and contamination by CheckM (*Parks et al., 2015*). CheckM was also used to assign taxa to these sequences using the *lineage* workflow. A bin was validated if its contamination rate fell under 10%. Bins with contamination levels higher than 10% were selected for recursive binning as previously described (*Baudry et al., 2019*). Briefly, the partition step was re-run 10 times on the corresponding sub-network of each contaminated bin, yielding smaller bins which were then re-evaluated using CheckM. The resulting bins above 500 kb were then also retained as MAGs. We used the Genome Taxonomy Database Toolkit with default parameters (GTDB-Tk) (*Chaumeil et al., 2020*) to validate taxonomic classifications consistent with the GTDB taxonomy and to define the novelty of the characterized MAGs.

## Protocols comparison

To compare the different protocols, we subsample the different libraries using (https://github.com/lh3/seqtk, *seqt, 2020*) and the same seed. We then fed the pipeline with the corresponding subsampled PE reads and their corresponding assemblies and process them as described previously in this section. Contact matrices and pie chart of the different ligation events were obtained using a home-made pipeline (https://github.com/koszullab/hicstuff, *Matthey-Doret, 2020*). Characterized MAGs for 3C, Hi-C, and eHi-C were also compared using the program LAST (http://last.cbrc.jp/).

## MAGs phylogentic tree

MAGs phylogenetic tree was obtained using CheckM and the *tree-qa* workflow (out format 3). Tree was pruned, visualized, and annotated using itol (https://itol.embl.de/) (*Letunic and Bork, 2016*).

## Phages contigs detection and classification

VirSorter v.1.0.3 (*Roux et al., 2015*) and VIBRANT v.1.0.1 (*Kieft et al., 2020*) were used to screen the 10 different assemblies. We removed the contigs annotated as prophages by the two programs and kept phages from all categories for VIRSorter. The remaining phages contigs were then classified as described in the *Figure 3—figure supplement 1*: (1) class A corresponding to contigs assigned to only one MAG after the iterative/recursive procedure (*Baudry et al., 2019*), (2) class B corresponding to contigs assigned to only one contaminated bin prior the recursive procedure and potentially corresponding to phages able to infect several species, (3) class C corresponding to contigs assigned to a bin smaller than 500 kb after the iterative procedure (but prior the recursive procedure).

Concerning contigs from category B, we calculated an association score of these contigs with the MAGs defined during the recursive procedure. This association score corresponds to the number of Louvain iterations where a contig was clustered with a MAG during the recursive process. Contigs exhibiting an association score ≥5 with only one preferential MAG were reintroduced in category A (*Figure 3—figure supplement 1*).

## Phages-host network analysis

Phage contigs were compared using two strategies. First, we used the software Mash v.2.0 to compute the distances between all phages contigs as well as between all characterized MAGS (*Ondov et al., 2016*), using Mash score = 1 as absence of homology and redefine it as a value of 0.3 as the maximum value after 1 is 0.28. Heatmap of the distance between all phages taken two by two and the sequence distance between their associated MAGs was smoothed using the kde2d() function of the MASS package of R (*Figure 3b*). Second, we used Blast to compare the different phage contigs from the 10 assemblies. All contigs sharing at least 70% of identity on 70% of their length were considered as belonging to the same genus.

We also used Blast to compare the different phage contigs. All contigs sharing at least 70% of identity on 70% of their length were considered as belonging to the same genus. We then compared the taxonomy/homology of their attributed MAGs and represented it using a circos plot.

CRISPR array in the characterized MAGs was identified using PILER-CR (*Edgar, 2007*). We then used Blast to screen spacers among the identified phages; 1252 robust hits between MAGs and phages were found (BLAST e-value <$10^{-5}$, bitscore ≥45), 387 of which in agreement with the host-phages assignments defined by 3D contacts. Most of the other hits involve MAGs and phages contigs from different samples (n = 781) but, in 84% of the cases (n = 660), these hits point to MAGs with same taxonomy.

## Phages contigs phylogenetic tree

Putative open reading frames (ORFs) inside phage contigs were detected using prodigal and the option *-meta* (*Hyatt et al., 2010*). A set of 77 protein hidden Markov model shared by dsDNA phages and defined previously (*Low et al., 2019*) was then used to screen the different predicted proteins (e-value <$10^{-3}$). For each contigs, the hit with the highest bit score was identified as the best hit and only the highest scoring sequence was selected. Only contigs with at least three markers among the 77 proteins were retained for further analysis. Markers were individually aligned using MUSCLE (*Edgar, 2004*), and individual multiple sequence alignment was trimmed using trimAl (*Capella-Gutiérrez et al., 2009*) at a threshold of 50%. Alignments were then concatenated using AMAS (*Borowiec, 2016*) by introducing gaps in positions where markers were absent from a contig. A phylogenetic tree was computed using IQ-TREE v.1.5.5 (*Nguyen et al., 2015*) and the following parameters: LG + GAMMA model and midpoint rooted, followed by 100 non-parametric bootstrap replicates. The tree was pruned, visualized, and annotated using itol (https://itol.embl.de/) (*Letunic and Bork, 2016*).

## Statistical analysis

R environment was used for all analyses (*R Development Core Team, 2020*). Statistical significance was set to p≤0.05. In *Figure 3b*, a chi-squared goodness-of-fit test was applied using the raw (unsmoothed) contingency table of the heatmap and cell equiprobability (1/number of cells) to define the theoretical contingency table. In *Figure 3c*, a Kruskal-Wallis test with continuity correction was used, considering that the different MAGs genus are ranked all along the circos plot.

## CrAss phages identification and phylogenetic tree

CrAss phages sequences from *Guerin et al., 2018* were downloaded and used as a database to search for homologous proteins against our set of contigs using prodigal and Blast (e-value $<10^{-5}$). Contigs with at least 10 significant hits were classified as potential crAss-like phages and were integrated in the phylogenetic tree. Phylogenetic tree was built as described in the upper section. We also integrated in the phylogenetic tree two crAss-like phages genomes of each genera from *Guerin et al., 2018*.

## Data availability

Sequence data (raw reads, assemblies) have been deposited in the NCBI Sequence Read Archive under the BioProject number PRJNA627086.

Code and additional files containing details on all the MAGs, bins, contigs, and phages can be found at the following address https://github.com/mmarbout/HGP-Hi-C; (*Marbouty, 2021*; copy archived at swh:1:rev:f2a185ed6638d445884177e319e831f88d67dba7).

## Acknowledgements

This research was supported by funding to RK from the European Research Council under the Horizon 2020 Program (ERC grant agreement 771813) and JPI-EC-AMR STARCS ANR-16-JPEC-0003–05. We thank all our colleagues from the laboratory, especially Pierrick Moreau, Lyam Baudry, and Cyril Matthey-Doret for discussions, feedback, and comments. We also thank MA Petit and A Laffitte for constructive comments on the data analysis and the manuscript, as well as our colleagues from the JPI STARCS consortium.

## Additional information

### Funding

| Funder | Grant reference number | Author |
| --- | --- | --- |
| European Research Council | 771813 | Romain Koszul |
| Agence Nationale de la Recherche | ANR-16-JPEC-0003-05 | Romain Koszul |

The funders had no role in study design, data collection and interpretation, or the decision to submit the work for publication.

### Author contributions

Martial Marbouty, Conceptualization, Formal analysis, Investigation, Methodology, Writing - original draft, Writing - review and editing; Agnès Thierry, Methodology; Gaël A Millot, Formal analysis; Romain Koszul, Conceptualization, Supervision, Funding acquisition, Investigation, Writing - original draft, Project administration, Writing - review and editing

### Author ORCIDs

Martial Marbouty (iD) https://orcid.org/0000-0002-1668-8423
Gaël A Millot (iD) https://orcid.org/0000-0002-0591-3509
Romain Koszul (iD) https://orcid.org/0000-0002-3086-1173

## Ethics

Human subjects: The work involved feces samples of healthy human individuals, stored in the Institut Pasteur biobanque (library). This research receives the ethical agreement n°N18 from Institut Pasteur (ICAReB), and through this process we dont need informed consent from the individual donors.

## Decision letter and Author response

Decision letter https://doi.org/10.7554/eLife.60608.sa1
Author response https://doi.org/10.7554/eLife.60608.sa2

# Additional files

## Supplementary files

• Supplementary file 1. Assembly statistics. Table indicating different metrics of the 10 sample and the resulting libraries and assemblies.

• Supplementary file 2. Mapping statistics. Table indicating different statistics on PE reads mapping and 3D ratio.

• Supplementary file 3. CrAss-like phages contigs. Table containing informations on the different detected crAss-like phage contigs.

• Supplementary file 4. Metagenomic assembled genomes (MAGs) data. Comma separated file describing the MAGs called in the study: sample, bin ID, bin size, bin mean GC%, bin mean coverage, taxonomy (seven levels), completion, contamination, contigs number, N50, mean contig size, longest contig, coding density.

• Supplementary file 5. Contigs data. Comma separated file describing all the binned contigs present in MAGs: sample, contig ID, contig size, contig coverage, contig GC%, associated bin.

• Supplementary file 6. Phages contigs data. Comma separated file describing all the phages' contigs associated to MAGs: sample, contig ID, associated bin.

• Transparent reporting form

## Data availability

Sequence data (raw reads, assemblies) have been deposited in the NCBI Sequence Read Archive under the BioProject number PRJNA627086. Code and additional data on MAGs, Bins, Contigs and Phages can be found at the following address https://github.com/mmarbout/HGP-Hi-C (copy archived at https://archive.softwareheritage.org/swh:1:rev:f2a185ed6638-d445884177e319e831f88d67dba7/).

The following datasets were generated:

| Author(s) | Year | Dataset title | Dataset URL | Database and Identifier |
|---|---|---|---|---|
| Koszul R, Marbouty M | 2020 | proximity ligation of human gut samples | https://www.ncbi.nlm.nih.gov/sra/SRX8404055 | NCBI Sequence Read Archive, SRX8404055 |
| Koszul R, Marbouty M | 2020 | proximity ligation of human gut samples | https://www.ncbi.nlm.nih.gov/sra/SRX8404054 | NCBI Sequence Read Archive, SRX8404054 |
| Marbouty M, Koszul R | 2020 | proximity ligation of human gut samples | https://www.ncbi.nlm.nih.gov/sra/SRX8404053 | NCBI Sequence Read Archive, SRX8404053 |
| Marbouty M, Koszul R | 2020 | proximity ligation of human gut samples | https://www.ncbi.nlm.nih.gov/sra/SRX8404052 | NCBI Sequence Read Archive, SRX8404052 |
| Marbouty M, Koszul R | 2020 | proximity ligation of human gut samples | https://www.ncbi.nlm.nih.gov/sra/SRX8404051 | NCBI Sequence Read Archive, SRX8404051 |
| Marbouty M, Koszul | 2020 | proximity ligation of human gut | https://www.ncbi.nlm. | NCBI Sequence Read |

| | | | | | |
|---|---|---|---|---|---|
| R | | | samples | nih.gov/sra/SRX8404050 | Archive, SRX8404050 |
| Koszul R, Marbouty M | 2020 | proximity ligation of human gut samples | https://www.ncbi.nlm.nih.gov/sra/SRX8404049 | NCBI Sequence Read Archive, SRX8404049 |
| Koszul R, Marbouty M | 2020 | proximity ligation of human gut samples | https://www.ncbi.nlm.nih.gov/sra/SRX8404048 | NCBI Sequence Read Archive, SRX8404048 |
| Marbouty M, Koszul R | 2020 | proximity ligation of human gut samples | https://www.ncbi.nlm.nih.gov/sra/SRX8404047 | NCBI Sequence Read Archive, SRX8404047 |
| Koszul R, Marbouty M | 2020 | proximity ligation of human gut samples | https://www.ncbi.nlm.nih.gov/sra/SRX8404046 | NCBI Sequence Read Archive, SRX8404046 |
| Marbouty M, Koszul R | 2021 | proximity ligation of human gut samples 8015_metaHiC_hpaII | https://trace.ncbi.nlm.nih.gov/Traces/sra/?run=SRR13435229 | NCBI Sequence Read Archive, SRX9848788 |
| Marbouty M, Koszul R | 2021 | proximity ligation of human gut samples 8015_metaHiC_MluCI | https://trace.ncbi.nlm.nih.gov/Traces/sra/?run=SRR13435228 | NCBI Sequence Read Archive, SRX9848789 |
| Marbouty M, Koszul R | 2021 | proximity ligation of human gut samples 16016_metaHiC_hpaII | https://trace.ncbi.nlm.nih.gov/Traces/sra/?run=SRR13435231 | NCBI Sequence Read Archive, SRX9848786 |
| Marbouty M, Koszul R | 2021 | proximity ligation of human gut samples 16016_metaHiC_MluCI | https://trace.ncbi.nlm.nih.gov/Traces/sra/?run=SRR13435230 | NCBI Sequence Read Archive, SRX9848787 |
| Koszul R, Marbouty M | 2020 | human gut metagenome Metagenomic assembly | https://www.ncbi.nlm.nih.gov/bioproject/PRJNA627086 | NCBI BioProject, PRJNA627086 |

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
