## [Decision Letter]

**Acceptance summary:**

This exciting and rigorous study utilizes a modified HiC method to increase metagenome assembled genome (MAG)-phage contig connections and the resulting data supports phage modulation of bacterial populations within the gut microbiota. This work lays the foundation for metagenomic applications that will advance our understanding of how phages impact their hosts and thus the microbiota both in the gut and in microbiotas other than the gut. This work is broadly applicable to many fields of research where microbial communities are studied.

**Decision letter after peer review:**

Thank you for submitting your article "Phages-bacteria infection network of the healthy human gut" for consideration by *eLife*. Your article has been reviewed by three peer reviewers, including Breck A Duerkop as the Reviewing Editor and Reviewer #1, and the evaluation has been overseen by Wendy Garrett as the Senior Editor. The following individual involved in review of your submission has agreed to reveal their identity: Christopher Quince (Reviewer #2).

The reviewers have discussed the reviews with one another and the Reviewing Editor has drafted this decision to help you prepare a revised submission.

Summary:

In this study, the authors apply chromosome conformation capture (3C) library preparation methods to ten fecal samples from healthy individuals. 3C allows physically collocated DNA molecules to be identified. It is well established for resolving genomic structure but its application to metagenomes is more novel. Here, the authors focus on its application to link bacteriophages in the human microbiome to their hosts. They determine that about 25% of the phages are lytic, whereas ~50% of the phages are dormant lysogens. They use this methodology to identify human phages belonging to an emerging family of viruses, CrAss phages, greatly extending the predicted host range of these phages. Overall, this paper largely supports previously observed biology of intestinal phages. What makes this new and of broad interest is the ability to increase metagenome assembled genome (MAG)-phage contig connection using a modified eHiC method, coupled with some innovative computational analysis to correlate MAG replication to phage contig abundances that suggest phage modulation of bacterial populations within the microbiome. The techniques outlined here have the potential to provide profound understanding of how phages impact their hosts and thus the microbiota beyond just the human gut.

The revisions below come from three independent reviewers, each who deemed this work important and of high quality. However, there was consensus that revisions are needed to improve the work, including more rigorous statistical analyses and the inclusion of more scientific detail. This study came across like a proof of principle exploring the alternate 3C methods and their effectiveness in this context rather than dramatically novel science. Taking this into consideration, it was our collective opinion that the methods themselves are still sufficiently novel that even as a proof of principle this is an interesting study that could provide guidance to future larger scale studies utilizing these methods. Additionally, we commend the authors for establishing their tools on a GitHub repository. The readme document contained within was very helpful in describing the scripts as well as what was used in this manuscript. This resource will be valuable to the field as a whole.

Essential revisions:

1) A major concern is the number of replicates used to verify the superiority of the eHiC technique to that of the HiC or 3C method. Only one sample is used to make this conclusion. Ideally the author's would have performed this on a few more samples and then compared these to several replicates of just metaHiC and meta3C. The authors say they identified 81 high-quality genomes. Are these the same as what was identified via the metaHiC and meta3C? This is a critical result to support the utility of this developed assay. Since the authors have already published that 3C technology can make phage-bacteria connections, the real novelty here is that the eHiC method greatly increases the number of connections that can be made. However, this should be more rigorously proven.

2) From these data, a major observation is reinforced, that temperate phages seem to dominate the virome. If most phages are temperate then is Hi-C actually needed to make valid and in depth host connections?

3) The authors indicate that they verified their connections via CRISPR matching; however, they do not show the data. This is important to include for two reasons: 1) they indicate that it complements their current analysis and 2) CRISPR matching is often thought to be state of the field for making connections and it is implied that this method works better. Additionally, an explanation of why 15% validation with CRISPR is reasonable.

4) The association between MAG taxonomy and the phylogeny of their associated phages. This is presented graphically in Figure 3C by comparing the phage phylogeny with the order level assignment of a MAG. There does appear to be an association, but this needs to be quantified. This could be demonstrated with a permutation ANOVA of the phylogenetic distance between phages against their order assignment (using adonis in vegan for example). It could then be determined if this association breaks down at higher MAG taxonomic resolution. Similarly, the data in Figure 3B could be used in a Mantel test to determine strength of association between MAG distance and phage distance.

5) Why were the “category B” phages excluded from the analysis. The distribution of their contacts across MAGs would be a good sanity check for the methodology. Ignoring these results seems inappropriate.

6) Simply observing twofold higher coverage for a phage than the host is not definitive evidence that the phage is lytic since multiple factors influence sequence coverage, the phage could be multicopy, it could be near the origin of replication, or possess a sequence composition biased to higher coverage. The authors need to demonstrate that the phage sequences have significantly higher coverage than expected given these sources of variation. Since they are using growth estimation methods they could look at position in the genome and compare with non-viral contigs that are at a similar point. It would be helpful if the authors could pull out specific phage-host pairs as examples of what these patterns look like.

7) When discussing the novelty of MAGs, this should really be placed in the context of the recent large-scale binning surveys (e.g. Pasolli et al., 2019) which similarly observed most novel diversity in the Clostridia.

8) Was there any taxonomic signal in the number of phages associated with a MAG?

9) Without a cell harvesting step; intra cellular crosslinking, digestion and ligation; or a large ligation reaction volume, how is random proximity ligation and / or crosslinking of extracellular DNA avoided following a freeze thaw cycle of sample with no storage buffer? This may not be an issue as the data seems to suggest a small degree of contamination in the fused reads, yet this was not explicitly tested experimentally with mock communities. Having the molarity, copy number or even mass of DNA entering the ligation step would likely help as it is suspected it is quite dilute in the 1.1 ml volume.

10) Overall the manuscript indicates that two method derivatives are developed, but only meta-eHiC is discussed. The Materials and methods list the procedures for preparing the meta-Hi-C and meta-eHiC libraries so one can deduce the differences in the methods, but the Results should take a moment to explain these two derivatives. Is not developing them a result of this study? This would frame the results and interpretation of this work for the reader.

11) What is the correlation supposedly presented in Supplementary Figure 2, Results? A statistic is needed to support this claim.

12) For VIBRANT and VIrSorter results clarification is needed concerning uniqueness. Are those considered unique between the two tools or unique across samples (and possibly also between the two tools)? Also, were all category 1-3 VirSorter phages included or were just category 1 or category 1 and 2 (like VIBRANT does)? This detail should be included in the Materials and methods.

13) By looking just at the active phage population (Class A), one could speculate that they were lytic/infectious which may be reducing bacterial population numbers. But the ori/ter ratio is not really providing insight into their population spread rather just their replication rate. It is unclear how the authors draw their conclusion of an inverse correlation from Figure 4A, right panel. Is there a statistic to quantify this observation?

---

## [Author Response]

Essential revisions:1) A major concern is the number of replicates used to verify the superiority of the eHiC technique to that of the HiC or 3C method. Only one sample is used to make this conclusion. Ideally the author's would have performed this on a few more samples and then compared these to several replicates of just metaHiC and meta3C. The authors say they identified 81 high-quality genomes. Are these the same as what was identified via the metaHiC and meta3C? This is a critical result to support the utility of this developed assay. Since the authors have already published that 3C technology can make phage-bacteria connections, the real novelty here is that the eHiC method greatly increases the number of connections that can be made. However, this should be more rigorously proven.

The reviewers are correct to point that more “eHiC” experiments are needed to strengthen the robustness of the conclusions drawn from the improved protocol. In order to avoid multiple names of related methods, we first simplified the nomenclature and called the improved protocols: metaHiC protocol I and metaHiC protocol II. In the revised version, we performed new experiments using these metaHiC protocols and generated four new metaHiC libraries (corresponding to sample 8015 and 16016, each using HpaII and MluCI as restriction enzymes). These libraries confirmed that the metaHiC protocol II systematically recovers much more enriched 3D signal ratio and performs a better binning than the other protocols, with less raw reads. All new data were now included in the global analysis, which led us to update the whole Results section.

We now compare MAGs retrieved using the different protocols (Figure 1—figure supplement 2) and show that MAGs retrieved using either meta3C, metaHiC protocol I or metaHiC protocol II are highly concordant (at a threshold of 80% of identity).

Please note that because of restriction to lab access and the overall situation, generating more metaHiC libraries than the three we now present in the manuscript was not realistic. We hope the reviewers will be satisfied with the inclusion of these experiments, that in our opinion clearly demonstrate the improvement we brought to the approach. That the quality of the data is increased using this protocol is indirectly backed by a recent work on pure cultures of bacteria and archaea (Cockram et al., 2020), where we improved the yield of Hi-C experiments by bringing related modifications to the protocol.

“A comparison between the MAGs retrieved using either the meta3C, metaHiC protocol I or II datasets showed that all approaches are highly concordant (Figure 1—figure supplement 3). The meta3C protocol is therefore the least enriched in useful 3D signal, but this can be overcome either by a higher sequencing depth, or by using the metaHiC protocol II.”

2) From these data, a major observation is reinforced, that temperate phages seem to dominate the virome. If most phages are temperate then is Hi-C actually needed to make valid and in depth host connections?

Indeed, our data confirmed that temperate phages engaged in lysogenic cycle seem to dominate the virome. We agree that in this case Hi-C related approaches are not necessarily essential (or maybe one would say the most convenient way) to link them to their hosts. However, the interest of the approach relies primarily on three points. First, it is useful to assign virulent phages to their hosts, even though they don’t represent the majority of the phage population. Second, temperate phages can be in a lytic and replicative cycle, in which case Hi-C is valuable to characterize their host as well (or/and sort things out). Indeed, a significant fraction of the virome is actually cycling phages or multicopy phages (~5%) of the detected phage contigs exhibit a coverage ratio over 2, (n = 367). Finally, Hi-C related methods have proven highly efficient on single samples as they are not based on covariance analysis, and therefore their interests extend beyond gut microbiota investigation.

3) The authors indicate that they verified their connections via CRISPR matching; however, they do not show the data. This is important to include for two reasons: 1) they indicate that it complements their current analysis and 2) CRISPR matching is often thought to be state of the field for making connections and it is implied that this method works better. Additionally, an explanation of why 15% validation with CRISPR is reasonable.

CRISPR is indeed efficient at associating phages to host (Minot et al., 2013; Stern et al., 2012). However, CRISPR is not distributed in all bacteria species (it is estimated that ~40% of bacteria and ∼70% of archaea encode a CRISPR system; Edwards et al., 2015; Held et al., 2013). It is also well documented that the spacers in a CRISPR array are rapidly turned over in the environment (Minot et al., 2013; Pride et al., 2011). Consequently, CRISPR spacers don’t necessary match any known sequence. Therefore, although this approach is specific (few false positives) it is not very sensitive (many false negatives) (Edwards et al., 2015). It has been estimated that bacterium with the most similar CRISPR spacer is the correct host for ~15% of the phages and that bacterium with the highest number of CRISPR spacers is the correct host for ~21% of phages (Edwards et al., 2015). Although these numbers are likely to vary, this leaves room for complementary techniques aiming at the same objectives. To better contextualize our results, we have modified the text as follows:

“The network was further corroborated by looking for CRISPR spacer matches between MAGs and phages contigs using PILER-CR and Blast over the entire network (all individuals data mixed)(Edgar, 2007; Edwards et al., 2015). […] This suggests that these MAGs have already been in contact with these phages during their evolutionary history, but are not in the present individual.”

4) The association between MAG taxonomy and the phylogeny of their associated phages. This is presented graphically in Figure 3C by comparing the phage phylogeny with the order level assignment of a MAG. There does appear to be an association, but this needs to be quantified. This could be demonstrated with a permutation ANOVA of the phylogenetic distance between phages against their order assignment (using adonis in vegan for example). It could then be determined if this association breaks down at higher MAG taxonomic resolution. Similarly, the data in Figure 3B could be used in a Mantel test to determine strength of association between MAG distance and phage distance.

As suggested, we have performed statistical tests to support this association. Regarding more specifically the phage phylogenetic tree, we did a permutation test (Krukal-Wallis) by considering the order of the genus of the associated MAG as the observed order and obtained a p-value of 0.0132.

We added sentences in the Results section and the Materials and methods section to explain our tests.

“Phage contigs were then positioned in a phylogenetic tree based on a set of 77 specific markers of double stranded DNA phages (Low et al., 2019) (Materials and methods). […] The tree unveils clusters of closely related phages infecting the same bacterial genus (Kruskal-Wallis test, p-value = 0.0132) suggesting, again, that phages are specific to their hosts in human gut.”

“The R environment was used for all the analysis (R Core Team, 2020). […] In Figure 3C, a Kruskal-Wallis test with continuity correction was used, considering that the different MAG genus are ranked all along the circos plot.”

5) Why were the “category B” phages excluded from the analysis. The distribution of their contacts across MAGs would be a good sanity check for the methodology. Ignoring these results seems inappropriate.

Phage contigs encompassed in the category B are not unambiguously assigned to a MAG. Therefore, they could represent different types of phages like free phages or phages able to infect different but phylogenetically related MAGs.

We have now analyzed those sequences further. Most of them (2,096 out of 2,460) do not cluster at all with a MAG over the 10 iterations of the recursive partition procedure. Over the 364 remaining contigs, only 7 clustered with at least two different MAGs over the 10 iterations, while 260 exhibit an association score of ≥ 5 / 10 (i.e. they cluster at least 5 times with a MAG over the 10 iterations of the recursive partition procedure). Therefore, it appears that most phages contigs sequences from categories B may correspond to free phages, and that very few exhibit clustering with several MAGs, reinforcing the idea that phages are highly specific and infecting only one species. Consequently, we have integrated 260 contigs from category B (the ones that exhibit an association score ≥ 5 / 10 during the recursive procedure) into category A, and added a paragraph to explain these results and better explain this category. See also Figure 3—figure supplement 1 which has been modified.

“Three classes (A, B, C) of phages were defined (Figure 3—figure supplement 1; see also Materials and methods). […] These results suggest that the majority (6,763 out of 9,488, i.e. ~ 70%) of the phages present within the human gut are specific, infecting only one bacterial species (Figure 3A).”

6) Simply observing twofold higher coverage for a phage than the host is not definitive evidence that the phage is lytic since multiple factors influence sequence coverage, the phage could be multicopy, it could be near the origin of replication, or possess a sequence composition biased to higher coverage. The authors need to demonstrate that the phage sequences have significantly higher coverage than expected given these sources of variation. Since they are using growth estimation methods they could look at position in the genome and compare with non-viral contigs that are at a similar point. It would be helpful if the authors could pull out specific phage-host pairs as examples of what these patterns look like.

These are excellent suggestions to validate this point, as all the aforementioned factors would have an influence on the coverage (an inherent bias of metagenomic analysis). We thank the reviewers for raising this point which has allowed us to refine the analysis, and most likely removed some false positive events.

We further investigated the potential influence of GC content on the ratio. Plotting the coverage ratio of the phages contigs as a function of the mean GC content of these contigs, or as a function of the GC ratio (phages contigs / MAGs mean GC%), did not unveil a correlation between the ratio and GC content (Figure 4—figure supplement 1B).

Then, we investigated the influence of replication activity in the host by searching in each associated MAGs contigs encompassing the dnaA gene and likely to carry the origin of replication of the bacterial genome (n = 856 MAGs implicating 6239 phages contigs) (Emiola and Oh, 2018). As expected, “dnaA” contigs (dubbed oriMAG contigs) exhibit a mean read coverage ratio higher than the other contigs (mean = 1.37; SD = 0.51; max = 5.15). The coverage ratio analysis was then refined considering the phage vs. oriMAG contigs coverage ratio instead of the mean coverage of the MAGs (Figure 4Aa). We classified the phages into four categories based on their read coverage ratio with oriMAGs contigs (Figure 4A and B).

Phages displaying a ~1:1 ratio with their oriMAG contig (category 1, 0.5 < ratio < 2.0) represented the vast majority of the 6,239 phages-host pairs (n = 4,477; ~75%). They correspond likely to dormant (pro)phages, able to reactivate or not. This population may also encompass active phages that incidentally exhibit the same coverage of their host. For 1,395 phage-hosts pairs, the oriMAG contig was significantly more covered than the associated phage (category 2 – undefined). This profile is consistent with an abortive phage infection cycle, a pseudo – lysogenic cycle or even with prophages not present in the whole host population. The remaining 367 pairs (~5%) corresponded to phages with a higher coverage than their assigned oriMAG contig: category 3 corresponded to a ratio between 2 and 4 (n=291), and category 4 to a ratio ≥ 4 (n=76). These phages may be (category 3) or are probably (category 4) actively replicating, potentially impacting the ecosystem through lytic activities. Overall, these results support the conclusions of former and recent work suggesting that the major part of the phages found in human gut are temperate phages with few lytic activity (Džunková et al., 2019; Minot et al., 2011; Reyes et al., 2010).

Figure 4 and the corresponding Results section have been changed accordingly.

7) When discussing the novelty of MAGs, this should really be placed in the context of the recent large-scale binning surveys (e.g. Pasolli et al., 2019) which similarly observed most novel diversity in the Clostridia.

We have performed a new analysis regarding the novelty of our retrieved MAGs using the latest GTDB-Tk database (release 0.95, July 2020), and have added a sentence in light of the publication from Pasoli et al.

“The 715 MAGs were then compared to the latest genome references of the GTDB-Tk databases (release 0.95; Chaumeil et al., 2020). […] This relatively low number of newly recovered species, although still significant, suggests that the databases about the human gut microorganisms (at least those describing western population) are converging towards completion.”

8) Was there any taxonomic signal in the number of phages associated with a MAG?

We haven’t detected any specific taxonomic signal in the number of phages associated with specific clades, this is now indicated in the main text.

“On average, a MAG was associated with 6.9 phages contigs, but this rate presented high fluctuations (SD = 7.4), with a maximum of 45 contigs (MAG #9010_74_0; see Supplementary files 4 and 6). No correlation between this rate and taxonomic annotation was identified.”

9) Without a cell harvesting step; intra cellular crosslinking, digestion and ligation; or a large ligation reaction volume, how is random proximity ligation and / or crosslinking of extracellular DNA avoided following a freeze thaw cycle of sample with no storage buffer? This may not be an issue as the data seems to suggest a small degree of contamination in the fused reads, yet this was not explicitly tested experimentally with mock communities. Having the molarity, copy number or even mass of DNA entering the ligation step would likely help as it is suspected it is quite dilute in the 1.1 ml volume.

The influence of fixation on frozen samples has been tested in the lab using human gut samples (the same sample was fixed directly – fresh – or after a frozen period of 6 month). It appears that the variation in 3D signal resulting from the frozen step affects slightly the DNA topology (less self-interacting domains, less DNA loops detected, etc.) However, we never observed a significant increasing of the noise signal between species.

The molarity of DNA entering the ligation step for metagenomics samples is unfortunately not easy to define, even though we don’t think this will be an important limitation in applying the technique to other samples. This quantity is highly dependent of the response of the sample to the breakage step, the restriction enzymes, the richness of the raw sample… In extreme cases (stool samples from sick children) we noticed that the protocol fails at generating optimal results, but typically these cases are highly predictable from the nature of the sample.

Note that in the present protocol (similar to current Hi-C protocols) we isolate the insoluble fraction of the sample, containing mostly DNA-proteins complexes, prior to the ligation step (Gavrilov et al., 2013). If DNA escape from the thawing cells, it is likely to be discarded as well at this step, and it is unlikely that significant amounts of free DNA fragments are engaged in this step. Moreover, most of the protocols published lastly on pure culture have significantly reduced the volume of the ligation step without any loss of contact map resolution or increasing of noise signal (Helmsauer et al., 2020; Rao et al., 2014). Therefore, we think it is reasonable to assume this step is not a limitation.

We nevertheless added a warning in the Materials and methods to raise awareness about this potential limitation, in some samples.

10) Overall the manuscript indicates that two method derivatives are developed, but only meta-eHiC is discussed. The Materials and methods list the procedures for preparing the meta-Hi-C and meta-eHiC libraries so one can deduce the differences in the methods, but the Results should take a moment to explain these two derivatives. Is not developing them a result of this study? This would frame the results and interpretation of this work for the reader.

We actually tried to “lighten” the technical part of the manuscript, simplifying the nomenclature, etc. We have also brought more details in the Materials and methods section. We hope that the manuscript reads more easily now with respect to this question, and that the interest of the derivative described here is now obvious.

11) What is the correlation supposedly presented in Supplementary Figure 2, Results? A statistic is needed to support this claim.

We agree that the correlation is not obvious and after reflecting and thinking further about it, it appeared to us that it is difficult to compare the different libraries regarding their noise signal. We have, therefore, removed the corresponding sentence and figure.

12) For VIBRANT and VIrSorter results clarification is needed concerning uniqueness. Are those considered unique between the two tools or unique across samples (and possibly also between the two tools)? Also, were all category 1-3 VirSorter phages included or were just category 1 or category 1 and 2 (like VIBRANT does)? This detail should be included in the Materials and methods.

We include for each sample all the phages detected by VirSorter and VIBRANT (all contigs identified as phage by one of these programs is retained for downstream analysis). As we wanted to compare phages and their characterized hosts between samples, we have not removed identical phages between samples. Regarding VirSorter, we have included all the phages (not prophages) from category 1 to 3. We have added the details in the Materials and methods section.

13) By looking just at the active phage population (Class A), one could speculate that they were lytic/infectious which may be reducing bacterial population numbers. But the ori/ter ratio is not really providing insight into their population spread rather just their replication rate. It is unclear how the authors draw their conclusion of an inverse correlation from Figure 4A, right panel. Is there a statistic to quantify this observation?

Actually, class A phages were not characterized as lytic/infectious but as non-ambiguously assigned phages. We have tried to be clearer in the main text. Within the Class A, phages exhibiting a read coverage ratio over 2 with respect to their assigned MAGs were suspected to be actively replicating phages and potentially active actors of bacterial population behavior and growth.

We agree with reviewers that the conclusion of an inverse correlation from Figure 4A is wrong and we did not have a metric for that. We thank them to point at this problem, since we could not detect any correlation (or anticorrelation) between bacterial growth rate and phages activity. We have also explored a possible correlation between phages ratio and MAGs abondance without success.

“To further question the influence of active phages, we investigated MAGs abundance and growth rate using GRID. […] No significant correlation was detected between the presence of active phages and the abundance/growth rate of their corresponding hosts, as assessed by this analysis (Figure 4C and D).”